# The Promise of Digital Self-Management: A Reflection about the Effects of Patient-Targeted e-Health Tools on Self-Management and Wellbeing

**DOI:** 10.3390/ijerph19031360

**Published:** 2022-01-26

**Authors:** Josefien van Olmen

**Affiliations:** Department of Family Medicine and Population Health, University of Antwerp, 2000 Antwerpen, Belgium; josefien.vanolmen@uantwerpen.be

**Keywords:** e-Health, self-management, well-being, chronic diseases, patient-provider interaction

## Abstract

Increasingly, people have direct access to e-Health resources such as health information on the Internet, personal health portals, and wearable self-management applications, which have the potential to reinforce the simultaneously growing focus on self-management and wellbeing. To examine these relationships, we searched using keywords self-management, patient-targeting e-Health tools, and health as wellbeing. Direct access to the health information on the Internet or diagnostic apps on a smartphone can help people to self-manage health issues, but also leads to uncertainty, stress, and avoidance. Uncertainties relate to the quality of information and to use and misuse of information. Most self-management support programs focus on medical management. The relationship between self-management and wellbeing is not straightforward. While the influence of stress and negative social emotions on self-management is recognized as an important cause of the negative spiral, empirical research on this topic is limited to health literacy studies. Evidence on health apps showed positive effects on specific actions and symptoms and potential for increasing awareness and ownership by people. Effects on more complex behaviors such as participation cannot be established. This review discovers relatively unknown and understudied angles and perspectives about the relationship between e-Health, self-management, and wellbeing.

## 1. Introduction

Doctors have long been the gateway to diagnosis and health care, since they have privileged access to diagnostic tools and medical knowledge [1]. Increasingly, however, people have direct access to e-Health resources, including: abundant health information on the internet from Dr. Google, medical guidelines, and online courses [2]; access to their personal and health care data via personal health portals [3]; and e-Health monitoring and self-management applications on their smartphones and other wearables [4]. Up to 85% of people in the US search for online health information regularly [5].

This direct access to e-Health resources is especially relevant for self-management for people with chronic diseases. Especially for people with chronic diseases, self-management has gained importance as an essential part of care and support, both within health service organizations, as well as in communities. It is considered a promising concept that, with right guidance and support, can empower people to address damaging health behaviors and address context challenges for better lifestyles [6]. Indeed, there are many examples of self-management programs that have been implemented [7,8].

In theory, patient-targeted e-Health tools can allow people to support self-management and reach their own health-related goals contributing to wellbeing. Yet the discourse and early evidence about patient targeted e-Health and its effect on self-management is mixed. Most scientific evidence pertains to intervention studies of health care provider-designed interventions supporting disease management and behavior change. A 2013 systematic review found the strongest evidence of positive effects for mobile health interventions for behavior change such as smoking cessation [9]. A 2021 review showed moderate effects on the medication adherence [10]. The effects of patient-directed e-Health tools that have been developed outside study settings are more scattered. Nevertheless, beliefs in the potential benefits and effects are prominent among developers and businesses [11], evoking critical voices about the commercial stakes and new power asymmetries [12]. Sociologists have pointed to the negative sociocultural dimensions of people being expected to become digitally engaged into their own medical care and preventive health efforts [13]. Obstacles to positive effects include the difficulty of filtering information [14] and usability of health applications [15]. The scarcity of impact evaluations and the variety of qualitative studies about different effects of e-Health on self-management and wellbeing have motivated us to examine the mechanisms between e-Health interventions, self-management, and wellbeing. This paper explores the existing knowledge about the relationship between these concepts and reflects upon existing evidence and unknowns. This is relevant since the e-Health market is growing more rapidly than regulations can follow. At the same time, health policies and self-management strategies are scaled-up because they are assumed to contribute to more wellbeing. The danger of this acceleration is that unintended and potentially negative effects are not addressed, such as the digital divide, self-stigma, and concurring inequalities. This review aims helps to identify areas for priority research in this domain.

## 2. Materials and Methods

### 2.1. Conceptual Framework

The explorative nature of the research question was best suited for a narrative review. A sampling of publications was done through keywords in three conceptual categories: (a) self-management; (b) patient-targeted e-Health tools; (c) health as wellbeing (Figure 1), and through snowballing starting from the references lists. The three concepts are understood in the following ways. Patient-directed e-Health tools were defined as digital health information, apps, and tools designed for direct and stand-alone use by laypeople without any involvement by professional providers. Self-management was considered from a broad perspective and using the United States Institute of Medicine definition “the tasks that individuals must undertake to live with chronic conditions including medical management, role management and emotional management, and the confidence to deal with those” [16]. The concept wellbeing originates from the philosophical concept “eudaimonia”—or living well [17], and in this research it includes outcomes such as subjective vitality, higher quality relationships, sense of meaning, and better physical health indicators.

### 2.2. Study Selection

Keywords for category (a) included self-management; for (b) patient websites, wearables, digital self-management tools; for (c) health and wellbeing. Papers about the relationship between one concept and another or both of the other concepts were selected. Interventions that were designed by health care providers and embedded within a comprehensive care approach were not included. The inclusion criteria allowed assessment of all English published papers in the selected databases, including editorials, literature reviews, critical reflections, and intervention studies. There was no date restriction, because the concepts of self-management and health/wellbeing were well researched in the 1960s, while e-Health tools for patients were introduced with the development of smartphones in the new millennium. The selection of papers was based upon the information provided about the theoretical development and empirical evidence of one concept in relation with the other concept(s). Three databases were included: Web of Science was the most inclusive database for robust scientific papers from both the medical and social domain; medline for any additional references; Google Scholar for gray literature; and the Cochrane review database to collect synthesized evidence on these topics. Exclusion criteria were protocol papers, papers without full text available, papers not within the prime scope of interest.

## 3. Results

The combination of keywords resulted in 49 reviews in the Cochrane database, 661 publications in Web of Science, 158 in Google Scholar and 19 in Medline. A prisma chart visualizes the further selection process. Snowballing through reference lists and peer-suggestions enlarged the list (Figure 2). In the final step, 64 publications were included. Many papers focused on a specific kind of e-Health tool: health information websites [5,18,19] with or without an interactive component; health wearables [20,21,22,23,24] or apps for a particular health condition [22,25,26,27,28,29,30,31,32,33,34,35,36,37,38]. Through snowballing, papers that described digital systems from a larger perspective were identified [11,39]. Most empirical studies examined the relationship between apps and self-management, whereas those about websites and wearables were often observational studies. Relatively few papers were found about wellbeing and its relationship with self-management and e-Health.

### 3.1. The Relationship between Patient-Directed e-Health Tools and Self-Management

The pathways through which websites, wearables, and apps contribute to different aspects of self-management vary.

Digitalization has allowed information and communication channels to be versatile, and new platforms and other creative digital outlets allow patients to acquire knowledge and skills in many different ways, suited to their personal profile, at their preferred time and place [40]. This is very clear for websites and apps providing information and supporting the interpretation of signs and symptoms. Direct access to the health information on the Internet or to smartphone apps can help people to interpret symptoms, to anticipate diagnosis, deal with complaints, and thus self-manage the behavior related to health or to disease. Most e-Health applications are designed to raise awareness among people with a specific condition, track related health data, and stimulate self-care behavior [41]. Apps influencing specific behavior often do so in the short term. Positive effects of e-Health on disease-specific self-management outcomes have been shown for different types of conditions [37]. The largest amount of evidence has been collected for digital self-management for diabetes [42]. Examples of other innovations for other conditions are digital games that improve exercise and physical activity for people in the cardiac rehabilitation [43]. People with mental illness report that apps have positive effects on specific disease-related symptoms such as depression and anxiety [34] and that this leads to increased feelings of ownership. However, some symptoms such as mood management are hardly addressed by self-management support [44]. Complex symptoms and behaviors are more difficult to address through health apps. For instance, apps to relieve pain experience lack robust evidence of efficacy [24]. There is very limited evidence that health apps increase people’s participation in activities, for instance in the social of behavioral domain [38]. Overall, the evidence for health apps showed effects of e-Health on specific (targeted) actions and symptoms in the short term, and the potential for people with specific conditions to become more aware of their symptoms, contributing to ownership of their health.

On the other hand, e-Health can also lead to uncertainty, stress, false reassurance, distrust, and avoidance, creating negative effects on self-management behavior. Uncertainties relate to the quality of information and to use and misuse of information. It is difficult to assess the relevance, quality, and reliability of data on the internet [45]. Information on many websites is poorly recalled by users [19]. Self-management devices are not easy to use and manuals are not understandable for the average person [46]. The usability of mainstream wearable devices such as Apple Watch and Fitbit is unsatisfactory and customer loyalty is low [47]. This limits the potential of e-Health tools to support people in self-management. The digital divide can contribute to increasing health disparities, disempowering people. This links directly to the next relationship, between self-management and wellbeing.

### 3.2. The Relationship between Self-Management and Wellbeing

Self-management is mostly researched in the context of self-management *support*, which typically include information, educational, psychological, practical, and social support. Many digital and other programs focus on Lorig’s set of self-management skills—problem solving, decision making, resource utilization, forming a patient-health provider partnership and taking action [48]. Reviews of self-management in health care organizations [49] and in policies [50] show that most programs focus on medical management [48]. Yet, the rise of chronic diseases induced a shift to behavior change and lifestyle management, which in turn led to the recognition of behavior theories and motivation as important mechanisms for change [51]. Conceptual thinking about wellbeing and its relationship with self-management emerged in the early 20th century, but its empirical evaluation of wellbeing has been relatively limited, both in the number and scope of studies.

The influence of stress and negative social emotions on self-management is recognized by people themselves [52] as a cause of negative spiral which also affects their wellbeing [53]. Health care providers also recognize this problem [54]. Yet, empirical research on this relationship is limited to health literacy studies, such as the effects of people not being able to understand instructions [55].

The focus on self-management has developed concurrently with a broader vision of health, but the link between these concepts is complex. Ryan, Huta, and Deci formulated a theory on the relationship among motivation, self-regulation, and wellbeing (as the broader interpretation of health), stating that eudaimonia—or living well—can be obtained by (a) pursuing goals that are intrinsically valued; and by (b) processes that are characterized by autonomy and awareness [17] and that this eudaimonic life is associated with various wellbeing outcomes. Thus far, the evidence base for the assumption underlying many health and self-management strategies and policies—that if people strengthen their body and mental functions this will contribute to better positive health—is still weak.

Policymakers, health care providers, and patients share the opinion that good self-management implies autonomy, pro-activeness, and responsibility [56] and that it comes with moral obligations of individuals towards society and towards one’s social network. How does the moral obligation of being a good self-manager, and its expectations and responsibilities, affect people, especially those that cannot live up to it? Existing evidence points to vicious cycles: establishing behavior norms [57] risks inducing feelings of shame [53] and self-stigma. Guilt or self-stigma can decrease social support and lead to psychological distress [58], further deteriorating health [59] and inequalities. Individuals in disadvantaged positions make shame-inducing comparisons with others regarding their social position, leading to stress and anxiety [60]. Research in work environments shows that a focus on self-management results in pressure on employees [61,62]. The links among expectations, stress and social emotions, and wellbeing have been examined in particular settings such as the diabetes care [54] and the reading of instructions [55]. These studies indicate that shame is present among people who are not being able to perform such self-management tasks, that it is not recognized, and that it negatively influences disease management and wellbeing.

### 3.3. The Relationship between Patient-Directed e-Health Tools and Wellbeing

The variety of e-Health tools (websites, apps, wearables) and the diverse drivers for people to use them make it difficult to synthesize the between e-Health and wellbeing. For instance, people use smartwatches mostly for recreative purposes which might lead to satisfaction and potentially wellbeing. The addition of medical apps, such as heart rate monitors or anxiety reduction, has provided opportunities for people to gain insight into their condition and its mastering. People using specific apps report that their usage led to increased feelings of ownership [38].

Health-related websites are also e-Health tools since they provide medical information and/or offer the option of social interaction on chat platforms. Experiences of people participating in interactions on such health platforms are generally positive [63]. People visit such platforms to gather information, to seek support, or to disclose. Reported positive effects are relief, reassurance, and reduction of fear [64,65]. Creative expressions, fun elements, practical tips, and positivity can also take away fear and concerns [38]. Web-based communities [63] can mitigate loneliness, isolation, and negative emotions through direct access to self-management support and a community of peers, providing opportunities for expression and release. This is valuable especially to those who experience insufficient support networks in their off-line environment [66]. The lack of hierarchy and the option for anonymity on such platforms contribute to the positive experiences of users.

However, e-Health can negatively affect wellbeing because of the potentially upsetting content, stress on how to use apps or navigate sites, and the fear to lose control over who has access to and sees the information. Web resources and search engines have increased in quality allowing more person-centered information [14], but the vast amount of information remains difficult to filter [67], especially for people with fewer digital skills. For instance, older adults not using the Internet reported feeling intimidated and anxious with technology [18]. Data exploitation is another factor affecting wellbeing. The transfer of ownership of personal health data from the user to a (commercial) app provider who might use it in unexpected ways potentially disempowers people [68]. On a larger scale, this may affect trust in the health system and in governments and lead to a shared feeling of lack of protection and safety. An obvious example is the recent emergence of distrust towards COVID-19 management strategies and the popularity of conspiracy theories about the digital control [69].

## 4. Discussion

Through a narrative review, this paper explores the complex relationships among patient-targeted e-Health tools, self-management, and wellbeing, discovering relatively unknown and understudied angles and perspectives. Most studies have examined and found proof of, the effects between patient-targeted e-Health and awareness and specific health—behavior outcomes, being a part of self-management. The effects on behavior from a more complex perspective, for instance, participation and overall functioning, or emotion management, have not been established nor been unraveled. Observational studies and critical reflections point to the potential negative effects of expectations towards people to self-manage such as stress and shame, which in turn influence wellbeing.

This paper has limitations. The data collection has been non-exhaustive. the analysis has not been structured. The exploratory research question about the nature of the relationships justified the approach of starting from broad keywords and snowballing to discover the domain. Not all available studies about self-management apps have been included, but the area has been generally covered. The review allowed us to observe an underrepresentation of studies about e-Health tools such as websites and wearables, and the paucity of evidence about the link with wellbeing. The selection of papers was guided by the research question, but the absence of strict inclusion criteria reduces the reproducibility of this process. The critical synthesis of the existing evidence is the result of a personal reflection, which is subject to selection and interpretation bias.

Nevertheless, the review reveals important areas of research: impact of commercialized e-Health tools on wellbeing; the influence of emotions on self-management; determinants of self-management at individual and at collective levels, and its relationship with empowerment; and longitudinal studies on the influence of e-Health on complex behavior’s. The effects of the (partial) digitalization of health care relationships have been examined [70,71], but the effects of patient-directed information on the asymmetry in information [72,73] and on power and decision-making have been understudied. This calls for health system research about the effects of the digital transition at the ecosystem level.

The review urges us to reflect upon the developments in the health and health care system. e-Health is changing health care systems and the relationship between patients and health care professionals [74], the organization of health care [75], and the further transformation of the delivery model of health care [76]. These innovations can disrupt power balances within the health system, shifting attention and resources towards citizens, for instance by redirecting part of health financing towards patient-targeting e-Health tools. This can facilitate the redesign of health systems and the reorganization of human capacity and resources, putting people’s empowerment, individually and collectively, at the center.

## 5. Conclusions

Patient-targeted e-Health tools can allow people to support self-management and reach their own health-related goals contributing to wellbeing. However, this is hampered by a lack of clarity and empirical evidence about the positive and negative influences on these relationships. Our initial exploration of these relationships warrants further research especially about the (mitigation of) negative effects. Health care resources are available via multiplying channels to many actors, and people are using e-Health with or without professional support. This fundamentally changes the traditional primary care-based health system model. It re-emphasizes the need for a better understanding of the meaning of self-management and wellbeing. It urges researchers, policymakers and people to reflect how we—at an individual and a societal level—relate to e-Health and further digitalization.

## Figures and Tables

**Figure 1 ijerph-19-01360-f001:**
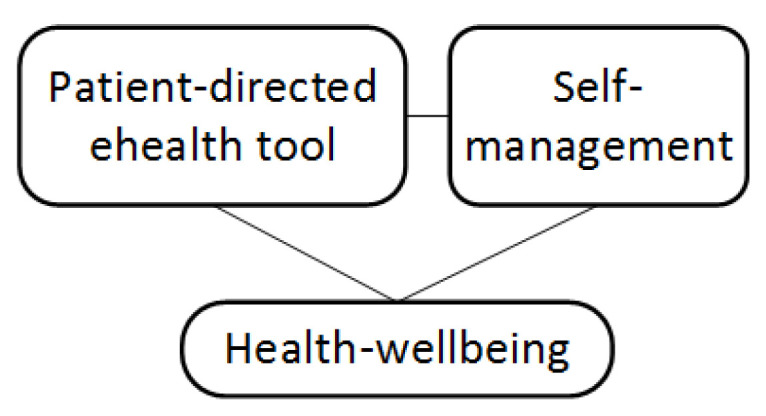
The relationships studied.

**Figure 2 ijerph-19-01360-f002:**
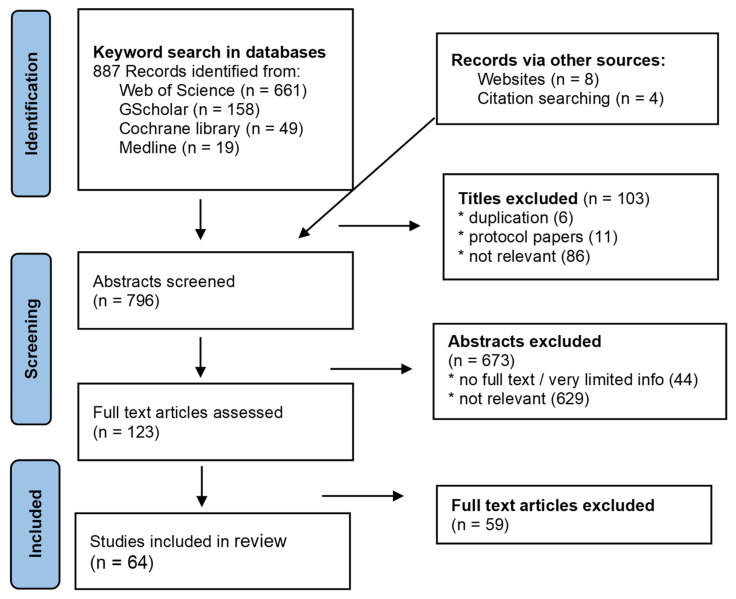
A prisma chart visualizing the selection process of papers for review.

## Data Availability

No new data were created or analyzed in this study. Data sharing is not applicable to this article.

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
