# Peer review of "The Promise of Digital Self-Management: A Reflection about the Effects of Patient-Targeted e-Health Tools on Self-Management and Wellbeing"

_ijerph, 2022, doi:10.3390/ijerph19031360_

Round 1

Reviewer 1 Report

The manuscript deals with the highly topical subject of digital self-management tools and investigates the relationship between these tools and individual self-management and wellbeing. The paper is clearly structured, pleasant to read and really adds an original contribution to the field of study. However, there are some aspects that need improvement from my point of view, before the paper is fit for publication.

I will address these points in the following and hope that they will help further improve the manuscript:

  1. The paper has a rather new and relevant perspective on the prominent phenomenon on e-health (including digital self-management tools) and its relationship with wellbeing and self-management. However, I am not convinced, the title is adequate to describe what has been done in the paper. As I understand it, the author has not only examined studies on self-management tools but also e-health tools in general (such as online communities or websites, which clearly are not used for self-management purposes). Hence, I would suggest changing the title and use the term “patient-directed e-health tools” instead. I would also recommend checking the paper thoroughly if these two different terms and concepts have always been differentiated enough.
  2. The introduction with the description of the theoretical (and empirical) background is rather short. This is probably due to a word limit, but I’d still like to read a bit more about the mixed results previous research has shown. Along these lines, the claim “Part of the reason for this is the lack of understanding about the mechanisms between eHealth interventions and self-management and wellbeing” on page 2 appears rather sudden and unsubstantiated. Where do you draw this conclusion from? Please elaborate a bit more.
  3. To increase methodological transparency, I would suggest including a figure, which illustrates the publication selection process (e.g. in the form of a PRISMA flow diagram). This figure would help make transparent why and at what point in the research project publications were excluded.
  4. While the concept of patient-directed e-health tools is well-defined, the paper would benefit from a detailed mention of the digital tools that were used in the interventions in the beginning of the findings. This information is currently splattered across the paper and hard to bring together.
  5. On page 3 in the findings, the author explains the concept of self-management with reference to Corbin and Strauss. In my view, this aspect actually belongs in the introduction, since it would help clarify the concept in the beginning and is not part of the interventions that had been studied by the author.
  6. Other than that, the findings are interesting to read and provide valuable insights into the state of research. The discussion, however, seems to be merely a summary of what had already been written in the findings. I would suggest including concrete scientific (and maybe also practical?) implications of what we can learn from the results of your review. Please also try to include limitations of your study.
  7. The author could think about making use of a professional proofreading service or finding another way to get the paper proofread. There are a couple of (in some cases recurring) mistakes (e.g., “on Internet” instead of “on the Internet”) that could be spotted and corrected with the help of a native speaker.

Author Response

Reviewer 1

The manuscript deals with the highly topical subject of digital self-management tools and investigates the relationship between these tools and individual self-management and wellbeing. The paper is clearly structured, pleasant to read and really adds an original contribution to the field of study. However, there are some aspects that need improvement from my point of view, before the paper is fit for publication.

I will address these points in the following and hope that they will help further improve the manuscript:

  1. The paper has a rather new and relevant perspective on the prominent phenomenon on e-health (including digital self-management tools) and its relationship with wellbeing and self-management. However, I am not convinced, the title is adequate to describe what has been done in the paper. As I understand it, the author has not only examined studies on self-management tools but also e-health tools in general (such as online communities or websites, which clearly are not used for self-management purposes). Hence, I would suggest changing the title and use the term “patient-directed e-health tools” instead. I would also recommend checking the paper thoroughly if these two different terms and concepts have always been differentiated enough.

Author: We have adapted the title accordingly. The new title is: The promise of digital self-management. A reflection about the effects of patient-targeted e-health tools on self-management and wellbeing. We have added a paragraph explaining the understanding of the three concepts including e-health. The term ‘self-management tools’ remains only in the keyword search section where it is appropriate.

  1. The introduction with the description of the theoretical (and empirical) background is rather short. This is probably due to a word limit, but I’d still like to read a bit more about the mixed results previous research has shown. Along these lines, the claim “Part of the reason for this is the lack of understanding about the mechanisms between eHealth interventions and self-management and wellbeing” on page 2 appears rather sudden and unsubstantiated. Where do you draw this conclusion from? Please elaborate a bit more.

Author: We have expanded the paragraph: we add more studies that initially crossed our path and led to the research question of the paper. Part of the answers are in the results, since the lack of studies led to us performing a narrative review. The new para reads as follows:

“In theory, patient-targeted eHealth tools can allow people to support self-management and reach their own health-related goals contributing to wellbeing. Yet the discourse and early evidence about patient target ehealth and its effect on self-management is mixed. Most scientific evidence pertains to intervention studies of health care provider designed interventions supporting disease management and behaviour change. A 2013 systematic review found best evidence for mobile health interventions for behaviour change such as smoking cessation [10]. A 2021 review showed moderate effects on medication adherence [11]. The effects of patient-directed ehealth tools that have been developed outside study settings are more scattered. The belief in its benefits and effects is prominent among developers and businesses [12], evoking critical voices about the commercial stakes and new power assymetries [13]. Sociologists have pointed to the negative sociocultural dimensions of people being expected to become digitally engaged into their own medical care and preventive health efforts . [14] Challenges to detecting positive effects include the difficulty of filtering health information [15] and usability of health applications [16]. The scarcity of impact evaluations and the variety of qualitative studies about different effects of ehealth to self-management and wellbeing have motivated us to examinethe understanding of the mechanisms between eHealth interventions and self-management and wellbeing. This paper explores the existing knowledge about the relationship between these concepts, and reflects upon the existing evidence and unknowns. This is relevant, since the eHealth market is growing more rapidly than regulations can follow. At the same time, health policies and self-management strategies are scaled-up founded upon the assumption that this will contribute to more wellbeing. The danger of this acceleration is that unintended and potentially negative effects are not addressed, such as the digital divide, self-stigma and concurring inequalities. This review aims helps to identify areas for priority research in this domain.”

  1. To increase methodological transparency, I would suggest including a figure, which illustrates the publication selection process (e.g. in the form of a PRISMA flow diagram). This figure would help make transparent why and at what point in the research project publications were excluded.

Author: I followed your suggestion. On Pg3, the following diagram is included

  1. While the concept of patient-directed e-health tools is well-defined, the paper would benefit from a detailed mention of the digital tools that were used in the interventions in the beginning of the findings. This information is currently splattered across the paper and hard to bring together.

Author: in the first paragraph of the result section we specified the ehealth tools that were found, with illustrative references.

The text inserted is:

Many papers focused on a specific kind of e-Health tool: health information websites [5], [19], [20] with or without an interactive component; health wearables [21]–[25] or apps for a particular health condition[26]–[31][32]–[35][36]–[38][23], [39]. Through snowballing, papers that described the digital systems from a larger perspective were identified [12], [40]. Most empirical studies examined the relationship between apps and self-management, whereas those about websites and wearables were often observational studies. Relatively few papers were found about wellbeing and the relationship with self-management and eHealth.”

  1. On page 3 in the findings, the author explains the concept of self-management with reference to Corbin and Strauss. In my view, this aspect actually belongs in the introduction, since it would help clarify the concept in the beginning and is not part of the interventions that had been studied by the author.

Author: The reviewer points to the need to elucidate better the concepts we use, including self-management, ehealth. We decided to make clarify those concepts in the methods section, also because the 2nd reviewer commented that the flow of the paper was could be improved by providing a clearer structure.

  1. Other than that, the findings are interesting to read and provide valuable insights into the state of research. The discussion, however, seems to be merely a summary of what had already been written in the findings. I would suggest including concrete scientific (and maybe also practical?) implications of what we can learn from the results of your review. Please also try to include limitations of your study.

Author: I have rewritten the discussion as follows:

Through a narrative review, this paper explores the complex relationships among patient-targeted eHealth tools, self-management and wellbeing, discovering relatively unknown and understudied angles and perspectives. Most studies have examined, and found proof of, effects between patient-targeted eHealth and awareness and specific health - behaviour outcomes, part of self-management. The effects on behaviour from a more complex perspective, for instance participation and overall functioning, or emotion management, have not been established nor been unraveled. Observational studies and critical reflections point to the potential negative effects of expectations towards people to self-manage such as stress and shame, which in turn influence wellbeing.

Study has limitations include the data collection and that theanalysis have not been exhaustive nor strictly structured. The exploratory research question about the nature of the relationships and a reflection about the existing evidence and unknowns justified the approach of starting with broad keywords and snowballing to discover the domain. Not all available studies about self-management apps have been included, but the area has been generally covered. The exploratory nature of the review allowed observation of the underrepresentation of studies about ehealth tools such as websites and wearables, and the paucity of evidence about the link with wellbeing. The selection of papers was guided by the research question and the absence of very strict criteria reduce the reproducibility of this process. The critical synthesis of the existing evidence is the result of a personal reflection, which is subject to selection and interpretation bias.

Nevertheless, the review reveals important areas of research: studies on the impact of commercialised e-health tools on wellbeing; the influence of emotions on self-management; the wider determinants of self-management at individual and at collective levels, and the conceptual relationship with empowerment, and longitudinal studies on the influence of ehealth on complex behaviours. The effects of the transition at ecosystem level also need more research. The effects of the (partial) digitalization of health care relationships have been examined [71], [72], but the effects of patient-directed information on the asymmetry in information [73] [74] and on power and decision-making have been understudied.

The review urges us to reflect upon the developments in the health and health care system. eHealth is changing health care systems and the relationship between patients and health care professionals [75], the organisation of health carey [77] and the further transformation of the delivery model of health care [76]. These innovations can disrupt power balances within the health system, shifting attention and resources towards citizens, for instance by redirecting part of health financing towards patient-targeting eHealth tools. This can facilitite the redesign of health systems and reorganise capacity and resources, putting people’s empowerment, individually and collectively, at the centre.

  1. The author could think about making use of a professional proofreading service or finding another way to get the paper proofread. There are a couple of (in some cases recurring) mistakes (e.g., “on Internet” instead of “on the Internet”) that could be spotted and corrected with the help of a native speaker.

Author: The paper was re-edited by a professional scientific editor.

Reviewer 2 Report

First, I want to say that proofreading of the manuscript by a native speaker of English is required as the article was difficult to follow due to the odd use of language and grammatical mistakes. I think that the material also needs to be better organised because, although the diagram on page 2 provides a focus, I often found myself losing track of the results. The presentation seems to be somewhat haphazard and the narration appears to ramble, although I think that the results are adequately documented. Another organising device that should be employed is a flowchart, which can be used to present the number of publications at each stage of the selection process. The author should investigate the use a PRISMA flow diagram for this. Finally, I wonder why Embase and Medline were not used to search for studies as these databases are commonly used in medicine? Also, why were conference abstracts excluded as they could be a rich source of information? Conference abstracts  are surely as worthy of consideration as grey literature. 

Author Response

Reviewer 2

  • First, I want to say that proofreading of the manuscript by a native speaker of English is required as the article was difficult to follow due to the odd use of language and grammatical mistakes.

Author: The paper was re-edited by a professional scientific editor.

  • I think that the material also needs to be better organised because, although the diagram on page 2 provides a focus, I often found myself losing track of the results. The presentation seems to be somewhat haphazard and the narration appears to ramble, although I think that the results are adequately documented.

Author: We did  a profound restructurual and re-editing. A number of changes are listed below:

- We added a paragraph on concepts in the methods, explaining the 3 concepts

- We rewrote the discussion, making it more distinct from results and contributing to a better flow

- We started the results section with an overview of the ehealth topics in the paper, providing a general introduction to the three sub result sections

- Repetitions were deleted, the narrative flow was improved.

  • Another organising device that should be employed is a flowchart, which can be used to present the number of publications at each stage of the selection process. The author should investigate the use a PRISMA flow diagram for this.

Author: We have indeed done so, on page 3, the flow diagram is inserted

  • Finally, I wonder why Embase and Medline were not used to search for studies as these databases are commonly used in medicine? Also, why were conference abstracts excluded as they could be a rich source of information? Conference abstracts are surely as worthy of consideration as grey literature.

Author:

- We included Medline in the screening. After deduplication 18 new articles, see prisma chart. Embase not available at our university. We went through all conference abstracts following the suggestion of the reviewer. Although many abstracts were excluded because of no full text available of not relevant (many about development, early phases), we did include 11 conference papers in full text screening – see prism chart -

Round 2

Reviewer 2 Report

There still needs to be improvement of the presentation in terms of use of the English language. For example, here are some quotes from the paper that are confusing or poorly worded, these are excerpts from a sentence that overall doesn't make sense:

'that with the right guidance and support can empower people'

'are scaled-up founded upon the assumption'

'results in complex and mixed links between'

'Study has limitations include the data collection'

There are many more examples of problems with the wording or leaving out 'the' in places, these are just some examples. 

However, the author has included a Prisma flow chart and improved the structure of the manuscript as well as used Medline as a resource and conference abstracts. However, some explanation should be given in the flow diagram as to the reasons why 59 full text articles were excluded. Also, the last sentence in the Materials and Methods section should be altered to reflect the fact that conference abstracts are now included. 

Author Response

There still needs to be improvement of the presentation in terms of use of the English language. For example, here are some quotes from the paper that are confusing or poorly worded, these are excerpts from a sentence that overall doesn't make sense:

'that with the right guidance and support can empower people'

'are scaled-up founded upon the assumption'

'results in complex and mixed links between'

'Study has limitations include the data collection'

There are many more examples of problems with the wording or leaving out 'the' in places, these are just some examples. 

Author response: We have gone through the text ourselves, and had another English speaking editor revise the text one more time. In track changes version the improvements are visible. If more editing is needed, I would be happy to pay MDPI for their paying editing services.

However, the author has included a Prisma flow chart and improved the structure of the manuscript as well as used Medline as a resource and conference abstracts. However, some explanation should be given in the flow diagram as to the reasons why 59 full text articles were excluded.

Author response: We have added the reasons in the flow diagram. These are: interventions designed by health care providers and/or part of a comprehensive approach, small-scale feasibility studies, low quality of analysis, no reporting of relevant outcomes, deviation from scope.

Also, the last sentence in the Materials and Methods section should be altered to reflect the fact that conference abstracts are now included. 

 Author response: We have deleted the words conference abstracts.

This manuscript is a resubmission of an earlier submission. The following is a list of the peer review reports and author responses from that submission.